# In Vitro Characterization of the Physical Interactions between the Long Noncoding RNA TERRA and the Telomeric Proteins TRF1 and TRF2

**DOI:** 10.3390/ijms231810463

**Published:** 2022-09-09

**Authors:** Patricia L. Abreu, Yong Woo Lee, Claus M. Azzalin

**Affiliations:** Instituto de Medicina Molecular João Lobo Antunes (iMM), Faculty of Medicine, University of Lisbon, 1649-028 Lisbon, Portugal

**Keywords:** telomeres, TRF1, TRF2, TERRA, RNA-protein interactions, R-loops, G-quadruplexes

## Abstract

RNA-protein interactions drive key cellular pathways such as protein translation, nuclear organization and genome stability maintenance. The human telomeric protein TRF2 binds to the long noncoding RNA TERRA through independent domains, including its N-terminal B domain. We previously demonstrated that TRF2 B domain binding to TERRA supports invasion of TERRA into telomeric double stranded DNA, leading to the formation of telomeric RNA:DNA hybrids. The other telomeric protein TRF1, which also binds to TERRA, suppresses this TRF2-associated activity by preventing TERRA-B domain interactions. Herein, we show that the binding of both TRF1 and TRF2 to TERRA depends on the ability of the latter to form G-quadruplex structures. Moreover, a cluster of arginines within the B domain is largely responsible for its binding to TERRA. On the other side, a patch of glutamates within the N-terminal A domain of TRF1 mainly accounts for the inhibition of TERRA-B domain complex formation. Finally, mouse TRF2 B domain binds to TERRA, similarly to its human counterpart, while mouse TRF1 A domain lacks the inhibitory activity. Our data shed further light on the complex crosstalk between telomeric proteins and RNAs and suggest a lack of functional conservation in mouse.

## 1. Introduction

Telomeres are nucleoprotein structures positioned at the natural ends of linear eukaryotic chromosomes, where they avoid unwanted activation of DNA repair machineries, telomeric DNA degradation and telomeric fusions [1,2]. Mammalian telomeres contain the multiprotein complex shelterin, which assembles through protein-DNA and protein-protein interactions [1,2]. In addition, telomeres contain the long noncoding RNA TERRA, which is transcribed from subtelomeric promoters towards the end of chromosomes by RNA polymerase II. Individual TERRA molecules comprise unique subtelomeric RNA sequences followed by (UUAGGG)n tracts of variable length [3,4,5]. A fraction of human TERRA associates with telomeric chromatin through RNA:DNA hybrids (telomeric R-loops or telR-loops) forming between the UUAGGG repeats and the template, C-rich telomeric DNA strand [3,6,7].

Several laboratories including ours demonstrated that the two human shelterin proteins TRF1 and TRF2 establish an intricate and functionally relevant crosstalk with TERRA. Recombinant TRF2 binds to TERRA oligonucleotides in vitro through different domains including its N-terminal glycine-arginine-rich (GAR) domain, a 44 amino acid long peptide also dubbed B domain due to its basic nature [2,7,8,9,10,11]. Recombinant TRF1 also binds to TERRA, through interactions occurring outside of its N-terminal, 66 amino acid long acidic (A) domain [7,10]. Confirming the validity of these in vitro observations, mammalian TRF1 and TRF2 physically associate with TERRA (UUAGGG)n sequences in human and mouse cells [10,12,13,14].

We showed that TRF2 stimulates TERRA invasion of telomeric double-stranded (ds) DNA in vitro, leading to the formation of telR-loops; this activity relies on TRF2 B domain, as a TRF2 variant lacking the B domain (TRF2ΔB) is unable to stimulate TERRA invasion [7]. Conversely, TRF1 A domain, while not being able to bind to TERRA, inhibits TERRA binding to TRF2 B domain and suppresses TRF2 ability to stimulate TERRA invasion [7]. In cells, TRF2 over-expression or replacement of TRF1 with a variant lacking the A domain (TRF1ΔA) causes aberrant accumulation of telR-loops which, in turn, provoke telomeric replication stress and rapid telomere loss [7]. While the functions associated to TRF2-induced telR-loops are still being studied, our data allowed us to propose that those telR-loops must be kept in check by TRF1 in order to avoid telomere instability.

Despite the importance of the complex interplay between TERRA, TRF proteins and telR-loops, the molecular and structural details of these interactions are still very limited. The UUAGGG tracts of TERRA can form G-quadruplexes (G4s) containing stacked Hoogsteen-bonded G-quartet motifs stabilized by monovalent cations including K+ [9,15]. Recombinant full-length TRF2 and its B domain alone bind to TERRA in a G4-dependent manner, as those interactions are stabilized by G4 stabilizers and, conversely, are lost when nucleotide substitutions impairing G4 formation are introduced in the RNA substrates [9,11,15]. Moreover, the substitution of 10 arginines (Rs) within the B domain into alanines (As) abolished the binding to TERRA, highlighting the importance of the GAR motif organization for RNA G4 binding [11]. To expand our understanding of the crosstalk between TERRA and TRF proteins, we set up an extensive array of biochemical assays using full-length and deleted TRF proteins or amino acid substitution variants in combination with different RNA oligonucleotides.

## 2. Results

### 2.1. RNA G-Quadruplexes Are Essential for Efficient Binding of TRF1 and TRF2 to Telomeric RNAs

Recombinant human TRF1 binds to TERRA in vitro through interactions occurring outside its A domain [7,10]. Indeed, TRF1 A domain does not interact with TERRA while TRF1∆A does almost as efficiently as full-length TRF1 [7]. Moreover, human TRF2 possesses several TERRA binding sites, also outside of the B domain, since variant proteins lacking the B domain bind efficiently to TERRA in vitro [7,10]. To address whether those additional binding sites within TRF1 and TRF2 recognize G4s, we performed RNA Electrophoretic Mobility Shift Assays (EMSAs) using N-terminally GST-tagged recombinant TRF1 and TRF2, as well as TRF1∆A, TRF2∆B and the B domain alone (Figure 1A and Appendix A). Proteins were incubated with radiolabeled RNA oligonucleotides of different sequences including: a TERRA-like sequence comprising 5 UUAGGG repeats, two variants unable to form G4s, where one or two Gs within each repeat were substituted by Cs (MUT1 and MUT2, respectively; Figure 1B), and two variants retaining the ability to form G4s where one or two Us were substituted by As (MUT3 and MUT4, Figure 1B). All proteins bound with comparable efficiencies to TERRA and G4-forming oligonucleotides, while they were essentially unable to shift RNAs not forming G4s (Figure 1C,D). Recombinant GST alone did not bind to any of the oligonucleotides used in the assays, confirming the specificity of the detected interactions (Figure 1C). These results are in agreement with data previously reported for TRF2 and the B domain [9,11] and show, for the first time, that TRF1 binding to TERRA also depends on the presence of G4s.

### 2.2. Positively Charged Arginines within TRF2 B Domain Are Essential for TERRA Binding

Two isoforms of human and mouse TRF2 B domains of different lengths have been reported [2,16], with the longest isoforms (long B, LB) containing 42 and 45 additional N-terminal amino acids compared to the shorter variants in human and mouse, respectively (Figure 2A). The alignment of long and short isoforms from both species (herein referred to as hB and mB for the shorter variants and hBL and mBL for the longer ones in human and mouse, respectively) reveals a high degree of conservation between the four peptides, with overall identity higher than 78% (Figure 2A). In particular, all R residues, which, due to their positive charge, contribute to the basic nature of the B domain, are conserved between human and mouse B domains. EMSAs with the four B peptides N-terminally tagged with GST (Appendix A) showed that they all bind with similar kinetics to TERRA oligonucleotides, with the longer isoforms presenting a slightly more efficient binding at lower concentrations (Figure 2B,C). We concluded that the functions depending on the interaction of TERRA with TRF2 B domain should be exerted similarly by the different TRF2 isoforms and continued our studies using the short B domains only.

To assess the individual contribution of the positively charged residues towards TERRA binding, we generated a series of mutants, where the 8 Rs clustered between positions 13 and 30 of the short human B domain were substituted with the neutral amino acid alanine (A) (Figure 3A and Appendix A). We created single substitution mutants for each R (R13A, R21A, R25A, R30A, R17A, R18A, R27A and R28A), two double substitution mutants for adjacent Rs (R17/18A and R27/28A) and one mutant, where all Rs were substituted by As (R13-30A). EMSAs with a single protein concentration (80 nM) showed that single or double R to A substitutions only slightly reduce TERRA-protein interaction efficiencies, while the B variant R13-30A failed to bind TERRA (Figure 3B,C). EMSAs with a broader range of protein concentrations confirmed that single and double R to A substitutions have a limited impact on binding efficiency (Appendix A), while the BR13-30A variant completely lost its ability to bind to TERRA at the used concentrations (Figure 3D,E). Finally, we generated a B domain variant where the 8 Rs were substituted with positively charged lysines (BR13-30K; Appendix A). This variant had a strongly diminished affinity for TERRA, although some binding was still retained at higher concentrations (starting at 80 nM; Figure 3D,E).

### 2.3. TERRA Binding to the B Domain Is Essential for TRF2-Mediated TERRA Invasion

Using plasmid invasion assays developed to detect the formation of telomeric RNA:DNA hybrids in vitro (Figure 4A), we previously showed that recombinant TRF2 increases the amounts of hybrids generated by the invasion of TERRA oligonucleotides into telomeric dsDNA [7]. Based on these data, we proposed that TRF2 stimulates TERRA invasion. However, our experiments could not formally exclude that the detected increment in invaded plasmid species derived from the stabilization of RNA:DNA hybrids, rather than from the stimulation of their formation. To clarify this point, we performed pulse-chase plasmid invasion assays where a plasmid containing a 1.2 kb long telomeric repeat sequence was incubated with radiolabeled TERRA oligonucleotides in the presence or absence of recombinant TRF2 for 30 min, followed by incubations with an excess of cold TERRA oligonucleotides for 30, 60 or 120 min. After incubation with cold oligonucleotides, we observed a slight decrease in invaded plasmids in reactions devoid of TRF2, suggesting that hybrids are relatively stable (Figure 4B,C). In the presence of TRF2, invaded plasmids increased over-time in reactions lacking cold oligonucleotides, as expected, while they substantially decreased in reactions containing cold oligonucleotides (Figure 4B,C). This indicates that TRF2 does not detectably stabilize already formed RNA:DNA hybrids, supporting our previous conclusion that TRF2 facilitates TERRA invasion into telomeric dsDNA.

Next, we performed plasmid invasion assays using GST-tagged full-length TRF2 variants with B domains carrying the R13-30A or R13-30K substitutions (Appendix A). TRF2R13-30K retained some ability to induce invasion, while TRF2R13-30A completely lost it, behaving identically to TRF2∆B (Figure 4D,E). These results strengthen our previously proposed notion that TERRA binding to the B domain is essential for TRF2-mediated TERRA invasion [7]. To confirm that the analyzed radioactive signals corresponded to RNA:DNA hybrid-containing plasmids, we treated the reaction products with RNaseH, which degrades the RNA moiety of RNA:DNA duplexes. As expected, this treatment strongly reduced the invaded plasmid signals (Appendix A).

The ability of TRF2 to promote invasion requires binding to telomeric dsDNA [7], hence the defects in invasion associated to TRF2R13-30A and TRF2R13-30K could derive from inefficient binding to telomeric dsDNA. We performed DNA EMSAs using the different TRF2 proteins and the same DNA substrate cloned in the plasmid used for invasion assays. Both mutants showed binding to telomeric dsDNA comparable to that of TRF2, with the DNA substrate being completely displaced already by 1 nM protein (Figure 4F,G), a concentration much lower than the ones used in our invasion assays (lowest concentration: 20 nM; Figure 4D). Therefore, the differences observed in invasion efficiencies cannot be ascribed to inefficient binding of TRF2 variants to dsDNA and are most likely due to different efficiencies of TERRA binding. Overall, these data indicate that TRF2 promotes hybrid formation, rather than stabilizing pre-formed ones, and it does so in a reaction requiring the 8 Rs clustered within its B domain between positions 13 and 30.

### 2.4. A Negatively Charged Patch of Glutamates within TRF1 A Domain Largely Accounts for Its Inhibitory Effect on TERRA-B Domain Complex Formation

We previously reported that fusing the A domain of human TRF1 to the B domain of TRF2 is sufficient to inhibit the binding of the latter to TERRA oligonucleotides [7]. We, thereby, sought to understand the nature of this inhibitory mechanism. Human TRF1 A domain is a 66 amino acid long peptide of acidic nature due to the presence of several positively charged aspartate (D) and glutamate (E) residues (Figure 2D). 50% of these residues are found clustered within two acidic patches at positions 35–37 (acidic patch 1, AP1) and 55–63 (AP2) (Figure 2D). The alignment of human and mouse A domains (hA and mA, respectively) reveals a higher degree of divergence for these sequences than for human and mouse B domains, in particular within the AP2 (Figure 2D). Consistent with what we previously showed [7], recombinant human A and a fusion of human A and B (AB) did not bind to TERRA oligonucleotides. Similarly, recombinant mA did not bind to TERRA oligonucleotides. However, mA was not able to prevent TERRA binding to the mB domain when fused to it (protein mAB in Figure 2E,F and Appendix A). This suggests that human and mouse TRF1 might differentially regulate TRF2 interaction with TERRA, possibly due to the low level of conservation of the AP2.

We then generated two deletion mutants of human AB, one comprising only the first 50 amino acids of the A domain (mutant A1-50B, Figure 5A and Appendix A) and the other one only the last 16 (mutant A50-66B, Figure 5A and Appendix A). When used in EMSAs, the two mutants showed strikingly different behaviors. The A1-50B mutant was able to bind to TERRA oligonucleotides with an affinity of roughly 60% compared to the one of the B domain alone (Figure 5B,C). Conversely, the A50-66B mutant failed to bind TERRA, similarly to the original AB fusion (Figure 5B,C). These results indicate that the last sixteen amino acids (50–66) of human TRF1 A domain, which contain the AP2, largely account for its inhibitory effect on TERRA-B domain binding. Furthermore, given the lack of conservation between human and mouse AP2s, these results explain the inability of mA to inhibit TERRA-mB domain interactions.

Finally, to directly assess the contribution of the Es within AP2 towards binding inhibition, we generated two new substitution mutants in which the eight Es were substituted by As (AP2EAB) or by negatively charged aspartates (AP2EDB) (Figure 5A and Appendix A). While AP2EAB regained roughly 80% of binding to TERRA oligonucleotides at the highest protein concentration tested (320 nM), AP2EDB almost did not bind at all to the oligonucleotides (Figure 5B,C). Hence, TRF1 A domain inhibits TERRA binding to TRF2 B domain largely through negatively charged Es within AP2. It is important to notice that the A1-50B and the AP2EAB mutants did not bind to TERRA oligonucleotides as efficiently as the B domain alone (Figure 5C). Hence, other amino acids within the A domain, possibly within the AP1, might contribute to the inhibition of TERRA-B domain binding, although at a lower extent than the AP2.

## 3. Discussion

The functional relevance of TERRA interactions with human TRF2 B domain has already been clarified, as these interactions promote the formation of telR-loops [7]. It remains to be understood when those TRF2-stimulated telR-loops are needed; it is possible that they become essential when telomeres shorten and produce more TERRA, in order to induce telomere elongation through homology-directed repair, as it has been proposed for budding and fission yeasts [17,18,19,20]. The relevance of the binding of other domains of TRF1 and TRF2 to TERRA still needs to be explored; however, they might also support telomere stability for example by regulating shelterin assembly. We have demonstrated here that both TRF1 and TRF2 binding to TERRA UUAGGG repeats depends on the ability of the latter to form G4s. In this light, our data strengthen the notion that G4 structures play crucial roles in safeguarding genome integrity [21]. Interestingly, it was recently demonstrated that the DNA repair factor RAD51 also binds to TERRA and promotes the formation of telR-loops in vitro [22]. It is thus possible that TRF2 and RAD51 cooperate in regulating telR-loop formation, perhaps by associating with the same TERRA moieties. Biochemical assays similar to the ones employed here should help probe this assumption, for example, by determining whether RAD51 binding to TERRA also depends on G4 structures.

The interaction of TRF2 B domain with TERRA G4s was reported also by the Lieberman laboratory [11]. In the same report, the authors showed that the small molecule N-methyl mesoporphyrin IX (NMM) binds with relatively high affinity to TERRA G4s and disrupts the TERRA-TRF2 B domain complex. Moreover, when they treated cells with NMM, they observed features of compromised telomere stability, including DNA damage marks at telomeres, rapid telomere shortening and under-replicated fragile telomeres [11]. While all those features were ascribed to the disruption of TERRA-B domain complexes, it is also possible that some of them could derive from the disruption of other interactions between TERRA and TRF2 regions outside of the B domain or TRF1. It will be important to identify all TERRA binding sites present in both TRF1 and TRF2 and study the contribution of each of them towards telomere stability.

We have also defined, with unprecedented resolution, the molecular details of how TRF2 B domain binds to TERRA and TRF1 A domain inhibits this binding. As for TRF2 B domain, the 8 Rs clustered within the GAR domain are necessary for binding to TERRA. Replacing them with positively charged Ks (BR13-30K mutant) impairs TERRA binding, although not completely, and the corresponding TRF2 full-length protein (TRF2R13-30K mutant) is still partly able to induce telR-loop formation. This suggests that electrostatic interactions and other types of contacts, likely established through arginine side chains, support TERRA-B domain interactions and consequent telR-loop formation. Consistently, it has been shown that the multivalent guanidinium group of arginines establishes multiple interactions with the backbone phosphates of all RNA bases, thereby stabilizing RNA-protein complexes [23]. In contrast, the inhibitory activity associated to TRF1 A domain appears to largely, if not exclusively, depend on the charges associated to the eight Es clustered within the AP2; indeed, the negatively charged AP2EDB mutant inhibits TERRA-B domain interactions as efficiently as the original AB protein. Overall, it is evident that the contribution of charged amino acids is fundamental in establishing the regulated interplay between TERRA and TRF proteins. Given the negatively charged nature of TERRA RNA, we speculate that positively charged Rs attract TERRA to the B domain and negatively charged Es repel it from telomeres or shield TRF2 B domain through RNA mimicry [24]. Our model also implies that independent, negatively charged patches placed in proximity of TRF2 at telomeres could exert functions similar to the ones exerted by TRF1 A domain. Interestingly, human RAP1, another shelterin component that directly interacts with TRF2, also contains an acidic domain comprised between amino acids 214 and 304 [2] ending with a stretch of eight consecutive Es. Because this acidic domain is located immediately before the interface interacting with TRF2, it is highly possible that RAP1 also regulates TERRA-TRF2 complex and, in turn, telR-loop formation.

TERRA-TRF protein interactions, by regulating telR-loop formation and telomere stability, might protect cellular and tissue homeostasis and avoid disease conditions. A survey of cancer-associated mutations found in TRF1 and TRF2 at the cBioPortal for Cancer Genomics [25,26] reveals that none of them is located within TRF2 B domain, suggesting a strong counterselection against altered B domain variants. Conversely, several mutations leading to amino acid substitutions within TRF1 A domain can be identified. One of those mutations causes a E to K substitution within the AP1 (E36K, found in one lung adenocarcinoma and two cutaneous melanomas) and two of them cause E to K substitutions within the AP2 (E55K and E58K, both found in bladder urothelial carcinomas). The impact of these substitutions on the inhibitory activity of TRF1 A domain still needs to be tested; nonetheless, it is tempting to speculate that an impaired inhibitory activity might cause telomere instability and contribute to cancer etiology and/or development.

Finally, we show that TERRA-TRF2 B domain interactions are conserved in mouse, while the regulatory function of TRF1 A domain is not. This suggests that mouse TRF2 promotes telR-loop formation, while mouse TRF1 cannot suppress them. If that were the case, factors other than TRF1, RAP1, for example, could be involved in restricting TRF2-stimulated telR-loops. However, conditional deletion of TRF1 in mouse embryonic fibroblasts leads to the appearance of telomere free chromosome ends [27], a phenotype also induced by TRF1 depletion in human cells [7]. We previously showed that, in human cells, this defect is specifically averted by the A domain of TRF1 and derives from aberrant telR-loops, as it can be suppressed by over-expression of the RNA:DNA hybrid-specific nuclease RNaseH1 [7]. While it is not known whether the telomere free ends induced by deletion of mouse TRF1 are also an outcome of aberrant telR-loops, it is possible that mouse TRF1 has retained the ability to suppress TRF2-induced telR-loops through mechanisms different from the ones in human. Further studies will be necessary to clarify the evolutionary conservation of the interactions between TERRA and TRF proteins and their regulation.

## 4. Materials and Methods

### 4.1. Plasmids

Plasmids pGEX-4T1 (GE Healthcare, Chicago, IL, USA) containing cDNAs of the proteins of interest N-terminally fused in frame to a Glutathione S-transferase (GST) sequence were used for expression in bacteria cells. Plasmids encoding for TRF proteins, TRF1ΔA, TRF2ΔB, the wild-type A and B domains, as well as the fusion of the two domains, were previously generated in the laboratory [7]. All subsequent deletions and single or double amino acid substitutions were obtained using the Q5 site-directed mutagenesis kit (New England Biolabs, Ipswich, MA, USA). cDNAs for TRF2 variants R13-30K and R13-30A were synthesized at GeneScript (Piscataway, NJ, USA), and those for BL, mB, mBL, mA, AP2EAB and AP2EDB at Integrated DNA Technologies (IDT, Leuven, Belgium). These sequences were then cloned into the pGEX-4T1 vector. All plasmids generated were validated by sequencing. The pTel plasmid used in plasmid invasion assays was previously described [7].

### 4.2. GST Protein Purification

The pGEX-4T1 plasmids were transformed into competent BL21 cells. Cells were grown in 5 mL LB media overnight at 37 °C and then 250 µL of those cultures were inoculated into 50 mL of fresh LB media. Cells were grown at 37 °C for 2 h, after which protein expression was induced with 50 μM IPTG (Sigma Aldrich, Merck Inc., Darmstad, Germany) for 5 h at 30 °C. Cells were collected by centrifugation (4200× *g* for 15 min at 4 °C) and lysed by sonication in 20 mL of GST pulldown buffer (50 mM potassium phosphate buffer pH 7, 1% Triton X-100, 100 mM NaCl, 5 mM EDTA, 0.15 mM PMSF) for proteins with molecular weights (MW) up to 50 kDa or in in Protein lysis buffer (50 mM Tris pH 8, 600 mM NaCl, 10% Glycerol, 1% Tween-20, 5 mM β-Mercaptoethanol, 1 mM PMSF) for proteins with MW above 50 kDa. Sonication was performed using a MSE Soniprep 150 apparatus (Sanyo, Osaka, Japan); samples were subjected to 2 cycles of 3 min “ON”, at maximum power, and 2 min “OFF”, while being kept on ice. Sonicated samples were then centrifuged at 4200× *g* for 20 min at 4 °C. After centrifugation, lysates were incubated with Glutathione agarose beads (Sigma Aldrich, Merck Inc., Darmstad, Germany) overnight at 4 °C on a tube roller. Beads were then washed three times with 5 mL of GST pulldown buffer and once with 2 mL of GST pulldown buffer low Triton (50 mM potassium phosphate buffer pH 7, 0.1% Triton X-100, 100 mM NaCl, 5 mM EDTA). Bound proteins were eluted in 200 μL of GST elution buffer (50 mM potassium phosphate buffer pH 7, 5 mM EDTA, 100 mM NaCl, 5% glycerol, 25 mM glutathione) for 20 min at 4 °C. Protein concentration and purity were determined using the Bradford protein assay (BioRad laboratories Inc., Hercules, CA, USA) and BSA reference samples, followed by fractionation in polyacrylamide gels and staining with BlueSafe reagent (NZYTech, Lisbon, Portugal). Note that all protein concentrations refer to monomers.

### 4.3. Electrophoretic Mobility Shift Assays (EMSAs)

For RNA EMSAs, 5-repeats oligonucleotides were synthesized at IDT (Leuven, Belgium). Sequences were as follow: TERRA, 5′-(UUAGGG)5-3′, MUT1, 5′-(UUACGG)5-3′, MUT2, 5′-(UUACCG)5-3′, MUT3, 5′-(AUAGGG)5-3′, MUT4, 5′-(AAAGGG)5-3′. Additionally, 12-repeat TERRA oligonucleotide [5′-(UUAGGG)12-3′] was obtained by in vitro transcription of a plasmid containing the respective DNA sequence downstream of a T7 promoter, using the HiScribe T7 High Yield RNA Synthesis Kit (New England Biolabs, Ipswich, MA, USA), following the manufacturer’s instructions. For dsDNA EMSAs, a ds(TTAGGG) fragment was excised by restriction digestion from the pTel plasmid and contained approximately 1200 bp of telomeric repeats flanked by non-telomeric sequences of 48 (upstream) and 12 bp (downstream). RNA oligonucleotides and dsDNA fragments were 5′-end labeled with T4 polynucleotide kinase (New England Biolabs, Ipswich, MA, USA) and [γ32P]-ATP and then purified using the Oligo Clean and Concentrator Kit (Zymo Research, Irvine, CA, USA). For RNA EMSAs, recombinant proteins and oligos were incubated in 20 µL of EMSA buffer (50 mM HEPES pH 8, 1 mM DTT, 100 mM NaCl, 0.01% BSA, 2% glycerol) containing 50 ng/µL *E. coli* tRNAs (Sigma-Aldrich, Merck Inc., Darmstad, Germany) for 20 min on ice followed by 10 min at 25 °C. For dsDNA EMSAs, *E. coli* tRNAs were excluded from the reaction mix and incubation was carried out for 30 min at 25 °C. After incubation, 4 µL of 6× gel-loading buffer (30% glycerol, 0.3% bromophenol blue) were added to the reactions, which were then electrophoresed in agarose gels and pre-cooled 0.5× TBE. 2% agarose gels were used for EMSAs with 5-repeat RNA oligonucleotides, 1.5% for 12-repeat RNA oligonucleotides and 1% for dsDNA EMSAs. Gels were run at 70 V for 1 h at 4 °C and then dried and exposed to a phosphoimager screen. Radioactive signals were detected with a Typhoon FLA 9000 or an Amersham Typhoon IP imager (GE Healthcare, Chicago, IL, USA). ImageJ was used to quantify signals. The number of independent replicates (n) is indicated in figure legends.

### 4.4. Plasmid Invasion Assays

Plasmid invasion assays were performed adapting the protocol previously established in the laboratory [7]. pTel plasmid was purified using the NZY Miniprep Kit (NZYTech, Lisbon, Portugal) and re-purified through Phenol-Chloroform extraction to assure removal of RNase contaminations. RNA oligonucleotides were as for EMSAs. 12.5 ng of plasmid (0.15 nM final concentration) were incubated in 20 µL of Invasion buffer (50 mM Tris-HCL pH 8.3, 10 mM DTT, 75 mM KCl, 0.01% BSA, 2% glycerol) containing 4 U of RNaseOUT (Thermo Fisher Scientific, Waltham) for 20 min on ice followed by 10 min at 25 °C, in the presence or absence of recombinant proteins. Labeled oligonucleotides were then added to the reactions at 1 nM concentration and incubated for 30 min at 25 °C. When specified, 60 U of RNaseH (Takara, Kusatu, Shiga, Japan) were also added in the reactions. For pulse-chase experiments, 1 nM of labeled oligonucleotides (hot) were incubated with 80 nM recombinant TRF2 for 20 min on ice followed by 10 min at 25 °C. Then, 12.5 ng of plasmid were added and reactions incubated for 30 min at 25 °C. Finally, 50 nM of un-labeled oligonucleotides (cold) were added to the reactions and the chase proceeded for 30, 60 or 120 min at 25 °C. All reactions were stopped by adding 1% SDS and 6 µg of Proteinase K and incubating for 15 min at 30 °C. Then 6× gel-loading buffer was added to a final concentration of 0.5×. Reaction products were fractionated by electrophoresis in 0.8% agarose gels, in 0.5× TBE. Gels were run at 45 V for 1 h at 25 °C and then dried and exposed to a phosphoimager screen. The radioactive signal detection and analysis were as for EMSAs. The number of independent replicates (n) is indicated in figure legends.

### 4.5. Statistical Analysis

For direct comparison of the two groups, we employed a paired or unpaired two-tailed Student’s *t*-test using GraphPad Prism (version 8.4.3, GraphPad software, San Diego, CA, USA). The *p*-values and used tests are indicated in figure legends.

## Figures and Tables

**Figure 1 ijms-23-10463-f001:**
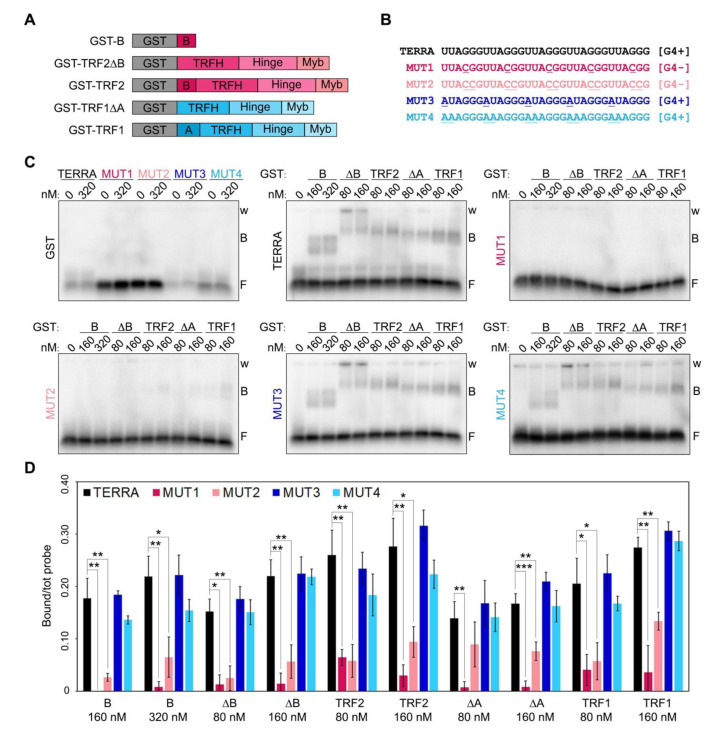
RNA G-quadruplexes are essential for efficient binding of TRF proteins to telomeric RNAs. (**A**) Schematic representation of the recombinant proteins used in EMSAs. (**B**) Sequences of the oligonucleotides used in EMSAs. G4+, oligonucleotides forming G-quadruplexes; G4-, oligonucleotides not forming G-quadruplexes; the nucleotides mutated relatively to the TERRA sequence are underlined. (**C**) EMSAs with different radiolabeled RNA oligonucleotides (0.25 nM) and the indicated concentrations of recombinant GST, GST-B, GST-TRF2∆B, GST-TRF2, GST-TRF1∆A and GST-TRF1. GST, glutathione S-transferase; B, bound probe; F, free probe; w, wells. (**D**) Bound oligonucleotides were quantified and graphed as a fraction of the total signal within each lane. Bars and error bars are means and SDs (n = 3 independent experiments except for GST for which n = 1). *p*-values were calculated with an unpaired two-tailed Student’s *t*-test. * *p* < 0.05, ** *p* < 0.01, *** *p* < 0.001.

**Figure 2 ijms-23-10463-f002:**
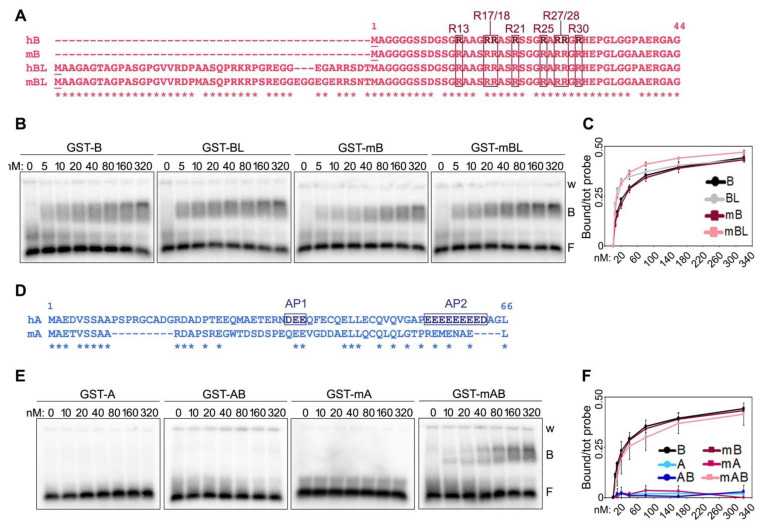
Comparison of human and mouse TRF N-terminal domains and their binding to TERRA. (**A**) Alignment of the short and long (L) isoforms of human (h) and mouse (m) TRF2 B domains. The asterisks (*) indicate conserved amino acids. The positions of the arginines comprised within the GAR domain are indicated. (**B**) EMSAs with radiolabeled (UUAGGG)5 TERRA oligonucleotides (0.25 nM) and increasing amounts of the indicated recombinant proteins. GST, glutathione S-transferase; B, bound probe; F, free probe; w, wells. (**C**) Bound oligonucleotides in experiments as in (**B**) were quantified and graphed as fraction of the total signal within each lane. Data points are means and error bars are SDs (n = 3 independent experiments). (**D**) Alignment of human and mouse TRF1 A domains. AP1 and AP2, acidic patches 1 and 2. (**E**) EMSAs performed as in (**B**) with the indicated recombinant proteins (**F**) Quantifications of experiments as in (**E**) (n = 3 independent experiments except for GST-A for which n = 1).

**Figure 3 ijms-23-10463-f003:**
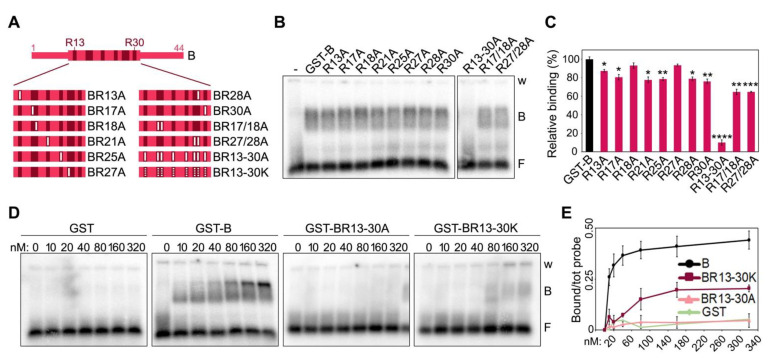
Positively charged arginines within TRF2 B domain are essential for TERRA binding. (**A**) Schematic representation of the human B domain substitution mutants used in EMSAs. Numbers indicate the position of arginine (R) residues within the B domain. Full white bars represent substitutions of Rs into alanine (A) residues; dotted white bars indicate substitutions of Rs into lysine (K) residues. (**B**) EMSAs with radiolabeled (UUAGGG)5 TERRA oligonucleotides (0.25 nM) and 80 nM of the indicated recombinant proteins. GST, glutathione S-transferase; B, bound probe; F, free probe; w, wells. (**C**) Bound oligonucleotides were quantified as a fraction of the total signal within each lane and graphed as the relative binding to GST-B. *p*-values were calculated with an unpaired two-tailed Student’s *t*-test. * *p* < 0.05, ** *p* < 0.01, *** *p* < 0.001, **** *p* < 0.0001. (**D**) EMSAs with radiolabeled (UUAGGG)5 TERRA oligonucleotides (0.25 nM) and increasing amounts of the indicated recombinant proteins. (**E**) Bound oligonucleotides were quantified and graphed as fraction of the total signal within each lane. Data points are means and error bars are SDs (n = 3 independent experiments except for GST for which n = 1).

**Figure 4 ijms-23-10463-f004:**
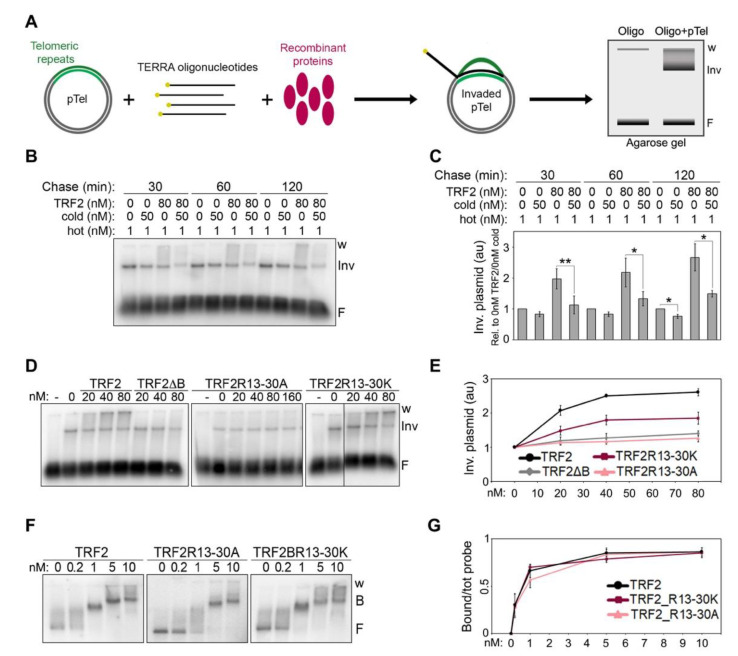
TERRA binding to the B domain is essential for TRF2-mediated TERRA invasion. (**A**) Schematic representation of the plasmid invasion assay. A plasmid containing a 1.2 kb long telomeric DNA array (pTel) is incubated with radiolabeled (yellow dots) TERRA oligonucleotides in the presence or absence of recombinant proteins and then electrophoresed in non-denaturing agarose gels. Oligonucleotide-invaded plasmids are visualized as slowly migrating species at the top of the gel, distinguishable from the free oligonucleotides (F) at the bottom of the gel. Note that for quantifications of invaded plasmids, the entire signal comprised between the well (w) and the lowest plasmid band (Inv) was utilized. (**B**) Pulse-chase plasmid invasion assays. The pTel plasmid (0.15 nM) was incubated with radiolabeled (UUAGGG)5 TERRA oligonucleotides (1 nM) in the presence or absence of TRF2 (80 nM); reaction products were then chased for the indicated times with 50 nM cold (UUAGGG)5 oligonucleotides. (**C**) Invaded plasmids were quantified as the fold increase compared with samples lacking protein and then graphed as relative to reactions containing no TRF2 and no cold oligo for each time-point. *p*-values were calculated with a paired two-tailed Student’s *t*-test. * *p* < 0.05, ** *p* < 0.01. (**D**) Plasmid and oligonucleotides as in (**B**) were incubated with increasing amounts of the indicated recombinant proteins. (**E**) Invaded plasmids were quantified and graphed as the fold increase compared with samples lacking proteins. (**F**) EMSAs with a radiolabeled dsDNA fragment containing a 1.2 kb long telomeric repeat sequence (0.03 nM) and increasing amounts of the indicated recombinant proteins. B, bound; F, free probe; w, wells. (**G**) Bound probes were quantified and graphed as fraction of the total signal within each lane. Bars and data points are means and error bars are SDs (n = 3 independent experiments).

**Figure 5 ijms-23-10463-f005:**
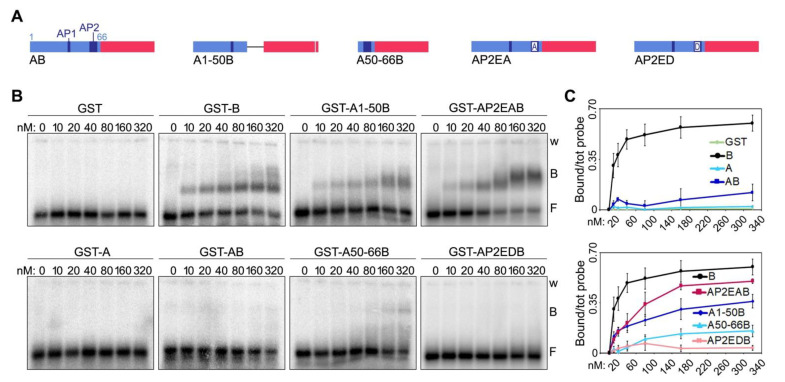
A negatively charged patch of glutamates within TRF1 A domain largely accounts for its inhibitory effect on TERRA-B domain complex formation. (**A**) Schematic representation of the recombinant proteins containing human A domain truncation and substitution mutants fused to the human B domain used in EMSAs. Numbers indicate the amino acid position within human A. (**B**) EMSAs with radiolabeled (UUAGGG)12 TERRA oligonucleotides (0.25 nM) and increasing amounts of the indicated recombinant proteins. B, bound; F, free probe; w, wells. (**C**) Bound oligonucleotides were quantified and graphed as a fraction of the total signal within each lane. Data points and error bars are means and SDs (n = 3 independent experiments).

## Data Availability

Not applicable.

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
