# Peer review of "In Vitro Characterization of the Physical Interactions between the Long Noncoding RNA TERRA and the Telomeric Proteins TRF1 and TRF2"

_ijms, 2022, doi:10.3390/ijms231810463_

Round 1
Reviewer 1 Report
The article by Abreu et al., entitled “Characterization of the physical interactions between the long noncoding RNA TERRA and the telomeric proteins TRF1 and TRF2”, submitted for publication in the International Jornal of Molecular Sciences, concerns the two human shelterin telomeric proteins TRF1 and TRF2 and their physical and functional interactions and crosstalk with TERRA, long noncoding RNA, which is transcribed from subtelomeric promoters towards the end of chromosomes by RNA polymerase II. Existing information on these mechanisms have been provide by this lab, as well as other laboratories. Here, the Azzalin lab, continuing its research on these mechanisms, further establishes additional and novel information on the TRF1-TRF2-TERRA intractions, with in particular the important role of the ability of TERRA to form G-quadruplexes in order to establish physical interactions with TRF1-TRF2. This new work is excellent, both in terms of scientific qualities and enormous quantities of convincing experimental data, and the interpretations and conclusions made by the authors are logical and well appropriate.
Comments:
1- Paragraph 2.3: For non-specialist readers, I think it would be useful to better explain the plasmid invasion assay, although it has already been published by the authors. Indeed, people know the EMSA, but the the plasmid invasion assay might be a little more sophisticated. In addition, the authors have performed this time additional controls using pulse-chase with cold TERRA oligonucleotides in order to distinguish between stabilization of already formed DNA::RNA hybrids and facilitation of TERRA invasion into double-stranded telomeric DNA. Given this, providing a little scheme explaining this experimental approach as well as its different interpretations might help the readers. I believe it would. Top of Figure 4 would be appropriate for this scheme.
2- I was a little puzzled for some time trying to understand the method for quantification in plasmid invasion assays, let’s say in Fig. 4C for instance. This is due to the fact that, in order to better understand the method, I went back to their 2018 Nat. Struct. Mol. Biol. paper, Fig1, in which the method of quantification was exactly the same as in the present paper, yet the signal between TRF2 and TRF1 was not as obvious in this previous paper as in the present one between TRF2 and TRF2DB. I believe this is due to the fact that in the present Fig. 4C there is some specific signal, not only in the lower “inv” band but also in the higher bands that are close to the wells (which is not the case in Fig. 1 of the 2018 paper). What might be misleading is in fact that the legend to Fig. 4D, which points out only to the “inv” bands, and the way the Fig. is labelled, one can think that only the lower band is to be taken into account, while, if I am correct, the whole signal between the lower “inv” band and the wells is pecific. Maybe this should be better explained, for instance stating that the whole signal is specific and that material localizing near, and within (?), the wells is real but is due to the presence of the proteins in addition to telDNA and RNA oligos.
3- This study have been well completed and provides us with a huge amount of novel in vitro elements, very likely susceptible of helping to better understand the TRF1-TRF2-TERRA interactions in vivo, while also exploiting their noted differences between humans and mouse. The experimental data are numerous, well designed and of high quality, as well as reasonably interpreted taking into account the numerous possibilities concerning the functions and roles of these interactions in the overall telomere biology.
Minor comments:
* Line 77: “preformed” should be “performed”.
* In Fig. 1A: the black letters “TRFH” and “A” are not well visible over the blue boxes.
* Line 100: “positive charge are contribute” should be “positive charge contribute”.
* Line 305; “unprecedent” should be “unprecedented”.
Reviewer 2 Report
In this manuscript from Abreu et al., in Azzalin laboratory demonstrated additional mechanistic details of the binding of TERRA RNA molecules to TRF2 and TRF1. The same lab has recently published in (https://www.nature.com/articles/s41594-017-0021-5) that the Basic domain of TRF2 and the acidic domain of TRF1 regulate, positively and negatively respectively, the RNA strand invasion to form telomeric DNA:RNA hybrid formation (also called telRloops). Respect to previous study, they show that the ability to form G4-quadruplex in TERRA molecules is key to drive the binding with both TRF2 (a fact that was already proposed) and TRF1 (novel observation). This occurs also in the absence of the A domain in TRF1 and B domain in TRF2, suggesting the presence of additional binding site for TERRA to these proteins. Indeed, mutant TERRA oligonucleotide sequences in which 2 G in each repeat have been substituted with C, which are unable to form G-quadruplex , do not interact with TRF2 wile mutant oligonucleotide which retain the ability to form G4-quadruplexes do.
In addition the show that a cluster of 8 positively charged Arginine residues in the GAR domain, are important to mediate TERRA-TRF2 binding and a patch of glutamate in the A domain of TRF1 mediate the inhibitory function of this telomeric factor on TRF2 B domain-TERRA interaction, thus modulating the extent of telomeric DNA:RNA hybrids formation. By using mutant form of TRF2 in which 8 positively charged Arginine have been mutated into alanine or positively charged lysins they are able to demonstrate that TERRA binding to TRF2 and telomeric DNA:RNA hybrid formation requires a set of at least 6 arginine residues in the domain (since single or double R-mutant still have an high affinity to TERRA) while the interaction with telomeric DNA is constant.
The study is well written and clear, despite the results looks a bit composites and the level of novelty is not very high.
I would recommend publication in IJMS after a short revision process that address the following point:
1) The study is entirely conducted in vitro. Would be reassuring to see some of key mechanism confirmed in vivo, in cells. As an example the key role of R residues in TRF2 B domain could be validated in vivo by simply expressing TRF2 R-to-A mutant isoform upon depletion of TRF1 or expression of a variant lacking the acidic domain and confirm the lack telR-loops formation and telomere loss. As a control the behavior of or R-to-K variant should be tested.
2) Many histograms in the figures do not present indication of which differences are statistically significant. This could help to appreciate what is more robust thus relevant, and should be added together with the indication of the statistical test used.
3) EMSA assays often present a lot of signal in the wells, when the RNA protein interaction is occurring. Would be nice to see if a more relaxed gel could reveal additional cooperatively interactions between TERRA and TRF2.
4) In the discussion it would be interesting to comment and elaborate more about other protein factors that have been show to promote telR-loops formation and how could support or compete with TRF2:TERRA mediated stimulation of these telomeric structure. Relevant examples are RAD51 and BRACA1 as described in this recent Nature article https://www.nature.com/articles/s41586-020-2815-6.
Round 2
Reviewer 2 Report
Despite the fact that no new results have been added to the paper to try to solve my concerns, the authors decided to changed the titled to highlight the pure in vitro nature of this study and decided to take away pieces of data still preliminary or unecessary.
In addition, they added statistical significance scores to the main graphs.
Therefore I have the impression that the present article, despite adding little new information to telomere biology, is now more more coherent.
Therefore I support publication in your journal